# Effect of pre-exposure prophylaxis on risky sexual behaviour of female sex workers in Dakar, Senegal: A randomised controlled trial

**Wally Toh**[1], **Aurélia Lépine**[2]*****, **Khady Gueye**[3], **Mame Mor Fall**[3], **Abdou Khoudia Diop**[3], **El Hadj Alioune Mbaye**[3], **Cheikh Tidiane Ndour**[3], **Owen O'Donnell**[1,4]

1 Erasmus School of Economics, Erasmus University Rotterdam, Rotterdam, The Netherlands, 2 Institute for Global Health, University College London, London, United Kingdom, 3 HIV/STI Division, Ministry of Health and Social Action of Senegal, Dakar, Senegal, 4 Erasmus School of Health Policy and Management, Erasmus University Rotterdam, Rotterdam, The Netherlands

* a.lepine@ucl.ac.uk

## Abstract

### Background

HIV prevention through pre-exposure prophylaxis (PrEP) may encourage riskier sexual behaviours that undermine the protection afforded by PrEP and generate negative spillovers through sexually transmitted infections (STIs). Tests for such risk compensatory behaviour in high-risk populations, such as female sex workers (FSWs), are lacking. This study aims to assess whether risk compensatory behaviours were observed among FSWs in Senegal after the rollout of PrEP.

### Methods and findings

In a randomised controlled trial with a Zelen design, we stratified FSWs in Dakar (Senegal) by self-reported sexual risk-taking and prior PrEP experience and randomly assigned them to immediate referral for oral PrEP (Treatment) from 7 September 2021 to end January 2022 or delayed PrEP referral (Control). We compared outcomes 3–8 months after the referral of the treatment group and before the referral of the control group. Primary outcomes were self-reported condom use with clients and perceived HIV/STI risks from sex with clients with and without a condom. The analysis is a modified intention-to-treat analysis. We estimated effects of PrEP referral as well as effects of oral PrEP use induced by randomised assignment to active PrEP referral. Out of 500 individuals randomised, 308 (61.6%) were included in the analysis (Treatment: 182/300 = 60.7%; Control: 126/200 = 63%). PrEP referral increased the probability of using oral PrEP by 34.5 percentage points (pp) (95% CI [25.4, 43.6]; $p < 0.001$). Estimated effects of PrEP referral and PrEP use on condom use with the last client were positive but not statistically significantly different from zero. PrEP referral was estimated to increase the probability of condom use

**Data availability statement:** All files are available from the UCL repository database DOI: https://rdr.ucl.ac.uk/articles/dataset/PrEP_randomised_controlled_trial_among_female_sex_workers_Senegal/28882685/1.

**Funding:** This study was supported by the MRC Public Health Intervention Development Scheme from UKRI, awarded to A.L., under grant number MR/T00262X/1. The study also received funding from the D.P. Hoijer Fonds, Erasmus Trustfonds, Erasmus University Rotterdam, The Netherlands, awarded to O.D. The full names and websites of the funders are: MRC Public Health Intervention Development Scheme (UKRI) Website: https://www.ukri.org D.P. Hoijer Fonds, Erasmus Trustfonds, Erasmus University Rotterdam Website: https://www.erasmustrustfonds.nl The sponsors and funders had no role in the study design, data collection and analysis, decision to publish, or preparation of the manuscript.

**Competing interests:** The authors have declared that no competing interests exist.

**Abbreviations:** 2SLS, two-stage least squares; ANCS, Alliance Nationale Contre le SIDA; CIs, confidence intervals; FSWs, female sex workers; ITT, intention-to-treat; MD, mean difference; MDE, minimum detectable effect; MoH, Ministry of Health and Social Action; OLS, ordinary least squares; pp, percentage points; PrEP, pre-exposure prophylaxis; PSA, prostate-specific antigen; RCT, randomised controlled trial; RD, risk difference; STIs, sexually transmitted infections.

with all of the last three clients by 11.0 pp (95% CI [0.8, 21.2]; $p = 0.034$). PrEP use was estimated to increase this probability by 25.8 pp (95% CI [5.2, 46.4]; $p = 0.014$). Main limitations were low power, high attrition, self-reported outcomes and a limited follow-up period.

## Conclusions

This study, conducted in one location, did not find evidence that PrEP referral or oral PrEP use increased self-reported risky sex behaviours of FSWs within 3–8 months. The robustness of this finding needs to be tested with larger cohorts followed for longer periods in other settings, and using survey instruments that allow further examination of whether PrEP users are more likely to overreport condom use.

## Trial Registration

ISRCTN—The UK's Clinical Study Registry, ISRCTN16445862 https://www.isrctn.com/ISRCTN16445862

## Author summary
### Why was this study done?

- Pre-exposure prophylaxis (PrEP) is becoming an essential part of HIV prevention among high-risk populations in low- and middle-income countries. However, compensatory risky behaviour may partially offset the protection PrEP gives against HIV and may increase the prevalence of other sexually transmitted diseases.

- Risk compensation may exhibit more strongly among female sex workers (FSWs) as condomless sex is better renumerated.

- A literature search in January 2024 revealed a lack of randomised controlled trials testing whether PrEP uptake impacts the prevalence of unprotected sex among FSWs. For men who have sex with men, the evidence was mixed, with more recent studies showing risk compensation.

### What did the researchers do and find?

- We randomised the timing of PrEP referral during its rollout in Dakar, Senegal and examined whether referral, and resulting use of oral PrEP, affected risky sexual behaviours, particularly condom use with clients.

- We found no evidence that either PrEP referral or oral PrEP use reduced condom use or increased measures of sexual risk-taking within three to eight months.

- This may be because users of oral PrEP reported believing that PrEP is more effective when combined with condom use.

### What do these findings mean?

- The evidence from this study, which was based on a moderately sized cohort in one location, does not lend support to concerns about a negative spillover from PrEP to more risky sex behaviours of FSWs.

- This positive result suggests that PrEP can deliver a net health benefit by reducing HIV risk without inducing an off-setting increase in STI risk.

- The robustness of this result needs to be tested with larger samples and longer follow-ups in other settings.

- The study's main limitations were low power, self-reported condom use measures, high sex work and survey attrition and a limited follow-up period.

## Introduction

When taken as prescribed, pre-exposure prophylaxis (PrEP) is highly effective in preventing HIV infection [1–2]. However, there is concern that this protection may encourage risky sexual behaviour—*risk compensation*—that partially offsets the preventive effect [2–7]. For example, those taking PrEP may reduce their use of condoms or increase their sexual partners, which could increase the prevalence of other sexually transmitted infections (STIs) that PrEP does not protect against.

There is a paucity of evidence on risk compensatory behaviour in response to PrEP [2]. Most research on the topic has focussed on men who have sex with men and has delivered mixed evidence [2,8–10]. Risk compensation seems more evident in more recent studies [8–10], which is consistent with an increasing behavioural response with growing awareness of PrEP's effectiveness [8–9].

Female sex workers (FSWs) can charge a higher price for condomless sex and other risky sexual acts. This financial incentive may increase the prevalence of risk-compensatory behaviour by FSWs taking PrEP [11]. However, there is a lack of robust evidence on the phenomenon in this population. Observational studies have shown mixed evidence [7,12–15] that is difficult to interpret because FSWs who take PrEP tend to engage in riskier sexual behaviours and, hence, may already be at higher risk of STIs prior to taking PrEP [15]. Qualitative research suggests that risk compensation may become more prevalent as FSWs become more convinced that PrEP is effective [16]. On the other hand, PrEP provision may also bring hard-to-reach populations, such as FSWs, into closer contact with the health system, increasing access to free condoms and lubricants, and improving opportunities for routine STI testing and sexual health counselling [3].

In Senegal, sex work is legal conditional on registration. Registered FSWs are required to visit a public health centre every month for health checks. Due to the strong stigma attached to sex work, many FSWs choose to remain unregistered. In 2015, HIV/AIDS prevalence among FSWs (6.6 percent) was nine times higher than in the overall population [17–18]. In 2021/2022, Senegal rolled out oral PrEP to targeted high-risk populations, including FSWs. Before the rollout, oral PrEP had been temporarily offered to only a limited number of FSWs in a 2015/16 PrEP Demonstration Project that was discontinued. This study aimed to use the randomised 2021/22 rollout of oral PrEP to FSWs in Senegal to obtain causal evidence on whether there is risk compensation in response to use of this effective HIV preventive medication in a high-risk population.

## Methods and findings

### Study design and participants

The study was a stratified randomised controlled trial (RCT) registered with ISRCTN (ISRCTN16445862). The study was approved by the National Ethics Committee for Research and Health Senegal (CNERS) (No. 00000031) and University

College London (UCL) Research Ethics Committee (17341/001). It is reported following the CONSORT (S1 Consort Guidelines) and SPIRIT guideline (S1 SPIRIT Guidelines) available as supplementary file. The trial protocol is provided as a supplementary file as well (S1 Trial Protocol).

Study participants were recruited from a list of FSWs who participated in a survey conducted between June and August 2020 in Dakar, Senegal. The survey was the third wave (after waves in 2015 and 2017) of a study following a cohort of FSWs at least 18 years old at entry. In each wave, the cohort was replenished with new participants who were recruited via snowball sampling by midwives at public health centres for registered FSWs and by peer facilitators for unregistered FSWs. By law, midwives are responsible for follow-up with registered FSWs.

We restricted study entry to survey participants who, based on information available in 2020, were potentially eligible for PrEP. That is, we excluded those who reported not doing sex work in 2020 and those who were recorded, in any wave, as having a public health system medical record showing they were HIV–positive. The 2020 survey was not a baseline for this study. It provided the base from which we recruited eligible study participants—approved by UCL Research Ethics Committee (17341/001)—and some data used for stratification and covariate controls. In each survey, consent was sought from participants for future contact.

The trial had a Zelen design [19]: informed consent was sought and obtained after randomisation. This design, including randomisation of the 2020 survey participants, was approved by the two ethics committees, without an explicit waiver of any requirement to obtain consent prior to random assignment for treatment. FSWs randomised to the treatment group were contacted and informed about the study's purpose, procedures, potential risks and benefits. If they consented to participate, they were given an appointment the following day for PrEP referral. Three to eight months later, all 2020 survey participants potentially eligible for PrEP (in 2020) who had been randomised to either the treatment group or the control group were invited to participate in an endline survey held from 11 April to 13 May 2022. Written informed consent was obtained from all participants at the beginning of this survey. The control group participants were never invited to participate in the trial's PrEP intervention. They were offered PrEP as part of its general rollout in Senegal after the trial. For the treatment group, the first consent for PrEP referral was 7 September 2021 and the last known consent was at the end of January 2022. For the control group, consent for PrEP referral was sought at the end of the endline survey.

## Randomisation and masking

Prior to contacting the 2020 survey participants who were potentially eligible for PrEP, on 10 October 2020, we randomised them to treatment and control groups (3:2) using a computer programme. Randomisation was stratified by prior experience with PrEP and sexual risk-taking, both of which were self-reported in the 2020 survey (Text A in S1 File). A participant could have had PrEP experience through the discontinued PrEP demonstration project [20]. Any behavioural response to the present study's intervention could have varied with prior PrEP exposure and propensity for risky sex behaviour, which was the reason for stratifying on these characteristics.

Enumerators were blind to treatment assignment when recording the primary outcomes—condom use and risk perceptions. After measurement of these outcomes, a question posed to only PrEP users revealed the treatment assignment of those participants. Treatment assignment remained blind for PrEP non-users. The midwives and peer facilitators who recruited FSWs to the endline survey were not blinded to the treatment assignment as they also helped with the referral for the PrEP intervention. They were not involved in collecting data.

## Procedures

The intervention was referral for oral PrEP. It was planned that rollout would begin soon after randomisation in October 2020, but it was delayed for various reasons. From early September 2021 to January 2022, midwives at public health centres and FSW peer facilitators actively contacted FSWs assigned to the treatment group and asked if they were interested in receiving PrEP. The midwives and FSW facilitators received comprehensive training from the HIV Division of

the Ministry of Health and Social Action (MoH) that developed the national PrEP protocol. The training included guidance on counselling, referral processes, and addressing potential barriers to PrEP uptake. Eligible participants were contacted by phone by a midwife or an FSW facilitator, who provided information about PrEP and its benefits. Participants who expressed interest and provided verbal consent were invited for PrEP eligibility screening on the following day at one of the designated referral sites.

The MoH and a non-governmental organisation—the Alliance Nationale Contre le SIDA (ANCS)—were the entities responsible for PrEP implementation, which included eligibility screening, PrEP distribution and follow-up visits. The MoH was mainly responsible for registered FSWs, who were screened and given PrEP at public health centres. ANCS mainly targeted unregistered FSWs and used community sites and mobile clinics for PrEP screening and distribution. PrEP eligibility was set by national guidelines that stipulated that oral PrEP be given to those (a) still doing sex work and (b) meeting medical criteria: (i) good liver function measured by creatinine level, (ii) HIV–negative, and (iii) not pregnant. Those identified as eligible were offered PrEP. The PrEP implementation partners were asked to defer PrEP screening of FSWs in the control group until after the endline survey. Treatment and control group participants who were still in sex work, not using PrEP and reported being interested in receiving it were referred for PrEP screening after the endline survey.

As the MoH and ANCS were responsible for PrEP implementation, for the treatment group, we have only incomplete information from administrative data on losses at each stage of recruitment and PrEP uptake. And there are no administrative data for the control group. Hence, the primary source of data is the endline survey, which was fielded from 11 April to 13 May 2022—3–8 months after the treatment group was referred for PrEP. We received no reported adverse events from the MoH and ANCS.

To recruit survey participants, midwives and FSW peer facilitators attempted to contact all FSWs randomised to either the treatment group or the control group. The interviews were carried out by trained enumerators at four public health centres, rented premises nearby, or quarters of trusted FSW facilitators. The survey asked about PrEP utilisation, preventive health and sexual behaviours.

Participants were asked: "Are you currently on PrEP to prevent HIV?". We coded those responding positively as 1 for the binary indicator *PrEP use*. Participants were also asked: "Have you started taking PrEP this year or last year?". We coded those responding positively as 1 for the indicator *Ever used PrEP in past year*. These self-reports of PrEP utilisation were cross-checked, as far as possible, with MoH and ANCS records, and by asking midwives and FSW facilitators to confirm whether each participant was effectively receiving PrEP at a health facility.

## Outcomes

The main objective was to determine whether PrEP affected condom use with clients. In the endline survey, we asked each participant to recall their last three clients and whether a condom had been used with each of them (Table A in S1 File). We created two binary primary outcomes: (a) 1 if a condom was used with the last client, and 0 otherwise; (b) 1 if a condom was used with all three clients, and 0 otherwise. These were multiple outcomes: a negative effect on either would be of concern. In supplementary analysis, we estimated effects on other prespecified outcomes: directly reported condom use with the penultimate client and measures of condom use elicited with two indirect methods that aimed to reduce reporting errors. In one of these—the double list experiment [21–22]—participants reported the number of correct statements from a list (Text B in S1 File). In both the treatment and control groups, participants were randomly assigned to one of two lists that differed by the inclusion or exclusion of a statement about condom use. The difference in the means of the reported number of correct statements between the two lists gave an estimate of condom use prevalence in the treatment group and in the control group. The other indirect elicitation instrument—the colorbox method [23–24]—obscured the participant's reported condom use from the survey enumerator (Text C in S1 File).

We used a 5-point Likert scale (1: Very unlikely,… 5: Very likely) to elicit perceptions of the risk of contracting HIV (STI) through sex with someone with HIV (STI) (a) if a condom were not used and (b) if a condom were used (Table A in S1

File). For each of HIV and STI, we created a binary outcome equal to 1 if the perceived risk was at or above the median response (= 5) in scenario (a): *High risk without condom*. Another binary outcome was 1 if the perceived risk was not above the median response (= 1) in scenario (b): *Low risk with condom*. We created a third outcome equal to 1 if both these indicators were equal to 1: *High risk without and low risk with condom*.

We included aspects of risky sex behaviours other than condom use as secondary outcomes. We asked participants to report the number of clients they had in a typical week and how many were regulars (as opposed to occasional clients). We calculated the proportion of regular clients. We asked each participant to report the average number of sex acts performed with each of their last three clients and whether they had oral sex and anal sex with any of these clients. Since a low proportion (<1%) reported anal sex, this outcome is not reported. Participants also reported their perceptions of the HIV risk of each of their last three clients (11-point Likert scale: 0 = no risk, 10 = very high risk). We created a binary outcome equal to 1 if a participant's maximum risk perception of the three reported was greater than the median (= 0). In addition, without pre-registration, we asked participants to report on an 11-point Likert scale how much risk they took in their recent sexual behaviours (0 = Limits risks, 10 = Likes to take risks). From this, we created a binary post-hoc secondary outcome equal to 1 if the response was above the median (= 1).

In the registered trial protocol, the specified primary outcomes were directly reported condom use with the last client, the penultimate client and all of the last five clients, as well as condom use elicited with the two indirect methods and perceptions of risks of contracting HIV and STI. Prior to fielding the endline survey—and therefore before the data lock—we decided to reduce the survey burden by asking (directly) about condom use with the last three clients, not the last five. We did not seek ethical board approval for this change because it involved collecting less data. In addition to those described above, pre-registered secondary outcomes included price charged to each of the last two clients, household expenditures and earnings from sex work in the last 30 days, food insecurity, self-reported STI symptoms with last two clients and mental health (Patient Health Questionnaire -9). After data collection and unlocking of the data, we decided not to use these outcomes in order to focus on risky sexual behaviours, which was further achieved by adding the post-hoc secondary outcome on subjective risk-taking. For consistency with the condom use outcomes, we used type of sex acts with the last three clients, not the last two as specified in the protocol.

We calculated the minimum detectable effect (MDE) on condom use with the last client given the fixed number of participants available from the 2020 survey. We assumed a drop-out rate of 10% from the original study size of 500 individuals due to survey and sex work attrition, a 5% PrEP ineligibility rate, a PrEP adoption rate of 82.4%, based on the rate achieved in the demonstration project [20], and a baseline condom use rate of 67.9%, based on the estimated prevalence in the 2020 survey elicited using list experiments. With 80% power and a two-sided two-sample proportion test (alpha = 0.05), for condom use with the last client, we calculated the intention-to-treat (ITT) MDE of PrEP referral as 13.1 percentage points (pp) and the MDE of PrEP use of 16.8 pp. We did not allow for covariates in the power calculations but adjusted for them in the analysis.

## Statistical analysis

We report percentages of the treatment group and control group who reported using PrEP at the time of the endline survey interview *(PrEP use)* and in the 12 months preceding that interview *(Ever used PrEP in past year)*, with binomial exact confidence intervals (CIs). We also provide whether the difference in PrEP usage is statistically significant between both groups using an unadjusted logistic regression. We used covariate-adjusted logistic regression to estimate the effect of assignment to PrEP referral – consisting of screening to establish eligibility and the offer of PrEP if eligible – on the probability of using PrEP *(PrEP use* and *Ever used PrEP in past year)*. In this analysis, and all others, we conducted inference with robust standard errors, which allowed for any form of unequal variances between groups [25]. CIs and p-values (*p*) were obtained from these standard errors.

We estimated modified intention-to-treat (ITT) effects of PrEP referral on the outcomes. Modification was due to randomisation prior to seeking consent and loss to follow-up because of attrition from sex work and the survey. We did not estimate effects of the offer of PrEP conditional on being deemed eligible since we had no information about eligibility in the control arm. We used adjusted logistic regression to estimate the risk difference (RD) in each primary (binary) outcome caused by PrEP referral. This is the estimated referral-induced change in the outcome probability for each participant (at observed covariates) averaged over all participants.

We estimated RDs in each primary outcome caused by *PrEP use* and *Ever used PrEP in past year*. To allow for non-random variation in PrEP use resulting from two-sided non-compliance, we used an encouragement design [26], with randomised assignment to active PrEP referral effectively used as an instrumental variable for PrEP use. For each outcome and each PrEP use measure, we used an adjusted recursive bivariate probit model for use (as a function of referral) and the respective outcome to estimate the RD as the change in the outcome probability induced by PrEP use averaged over all participants. We also calculated this RD relative to the predicted outcome probability if there were no PrEP use, which was obtained by averaging predicted outcomes from the estimated bivariate probit under the counterfactual of no PrEP use. In supplementary analysis, we treated the categorical risk perceptions reported on a 5-point Likert scale as linear measures, scaled 1–5, and used ordinary least squares (OLS) to estimate the mean difference (MD) in each of these outcomes caused by PrEP referral. We used two-stage least squares (2SLS)—with randomisation to PrEP referral as an instrument for PrEP use—to estimate the MD in each of these outcomes caused by each measure of PrEP use.

Covariate adjustment was done to (a) control for any treatment-control group imbalance that may have resulted from attrition, (b) increase precision, and (c) control for systematic non-compliance when estimating effects of PrEP use. The covariates used in all analyses were age (years), number of days in the seven preceding the endline interview that were within Ramadan—to account for the reduction in sex work during Ramadan—and 2020 values of marital status, FSW registration status, self-reported sexual risk taking, prior PrEP experience and the respective lagged outcome. In supplementary analysis, we estimated effects on the condom use outcomes without covariate adjustment and with entropy balancing [27] used to reweight and match, on moments of covariates, (a) the (post-attrition) cohort used for analysis to the pre-attrition cohort and (b) the treatment group to the control group in the analysis cohort.

We used adjusted logistic regression and bivariate probit to estimate RDs in binary secondary outcomes associated with PrEP referral and (each measure of) PrEP use, respectively. To estimate MDs in non-binary secondary outcomes associated with PrEP referral and use, we used OLS and 2SLS, respectively.

To assess whether PrEP users viewed PrEP and condoms as substitutable or complementary methods of HIV prevention, we asked only these participants to report on a 5-point Likert scale (1: Very unlikely, 5: Very likely) their perceptions of the risk of contracting HIV through sex with someone with HIV if they were to use (a) no PrEP and no condom, (b) no PrEP but a condom, (c) PrEP but no condom, and (d) both PrEP and a condom. We compared the perceived risks across these scenarios.

All statistical analyses were performed using R 4.1.2 and STATA 17.0

## Results

Out of 604 FSWs surveyed in 2020, 500 (82.8%) were assessed, based on 2020 data, as potentially eligible for PrEP and were randomised to active PrEP referral (treatment, *n* = 300) or delayed PrEP referral (control, *n* = 200) (Fig 1). Out of the 500 FSWs randomised, 308 (61.6%) were used in the analysis, with 95 (19.0%) excluded because they could not be traced or they refused to participate in the endline survey and 97 (19.4%) excluded because they reported not doing sex work at the time of the survey and, hence, sex work outcomes could not be elicited (Fig 1).

Table 1 shows characteristics, at the time of the 2020 survey, of the participants used in the analysis (*n* = 308). Overall, they had a mean age of 39 years, over half (54%; 167/308) had no schooling, almost four fifths (79%; 243/308) were divorced, separated or widowed, 45% (138/308) were registered FSWs, and a vast majority (97%, 298/308) reported

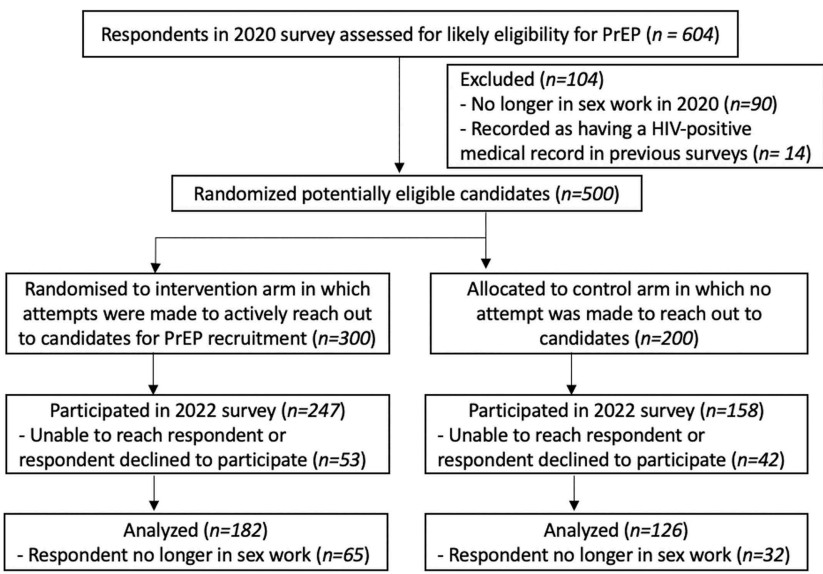

**Fig 1. Participant flow.** PrEP, pre-exposure prophylaxis.

using a condom with their last client. On average, participants had 6.5 clients in a typical week, 72% of their clients were regulars, and they derived 84% of their income from sex work.

In this analysis cohort, the treatment and control groups were balanced on baseline characteristics except for differences in marital status and client numbers (Table 1). A joint F-test did not reject the null hypothesis of no differences across the multiple characteristics ($p = 0.275$). There was balance in the pre-attrition cohort (Table B in S1 File). Selection into the analysis cohort ($n = 308$) from the pre-attrition cohort ($n = 500$) did not differ by treatment assignment and most baseline characteristics, with the exceptions of household indebtedness and one of the randomisation strata (Table C in S1 File).

Sociodemographic and sex work characteristics of the analysis cohort participants reported at endline (2022) were similar to those reported in the 2020 survey (Table D in S1 File). There was a 10 pp reduction in reported condom use with the last client and a fall in the proportion who reported sexual risk-taking.

Table 2 shows percentages of the control and treatment groups that, at endline, reported each measure of PrEP use. It also shows unadjusted RDs in these measures between the groups. We estimated that assignment to active PrEP referral increased the probability of reporting (a) PrEP use at endline by 34.5 pp (95% CI [25.4, 43.6]; $p < 0.001$) and (b) PrEP use in the past year by 39.1 pp (95% CI [29.4, 48.7]; $p < 0.001$).

Table 3 shows adjusted RDs in condom use and perceptions of HIV and STI risks by PrEP referral and the two measures of PrEP use (Table E in S1 File gives treatment and control group mean outcome probabilities). There are estimated increases in the probability of using a condom with the last client resulting from PrEP referral and each measure of PrEP use, but none of these estimates are statistically significant. PrEP referral was estimated to increase the probability of using a condom with the last three clients by 11.0 pp (95% CI [0.8 pp, 21.2 pp]; $p = 0.034$)—a 16.3% increase relative to the mean outcome probability without referral (67.5%) (Table E in S1 File). *PrEP use* was estimated to increase the probability of using a condom with the last three clients by 25.8 pp (95% CI [5.2, 46.4]; $p = 0.014$)—a 36.3% increase relative to the mean outcome probability without PrEP use (71.1%) (Table F in S1 File). The RD in this outcome by Ever used PrEP in the past year is similar.

Estimates of the RDs in condom use are similar without adjustment for covariates (Table G in S1 File) and with reweighting to match, on moments of covariates, (a) the analysis cohort participants to the pre-attrition cohort and (b) the treatment group to the control group in the analysis cohort (Table H in S1 File). The estimated RDs in condom use with

PLOS Medicine

**Table 1. Analysis cohort characteristics at baseline.**

| | Overall (n = 308) | Control (C) (n = 126) | Treatment (T) (n = 182) | p(H₀: C = T) (n = 308) |
|---|---|---|---|---|
| **Sociodemographics** | | | | |
| Age (years) | 38.9 (9.4) | 39.2 (9.2) | 38.8 (9.5) | 0.700 |
| Ever went to school | | | | 0.442 |
| Yes | 141 (46%) | 61 (48%) | 80 (44%) | |
| No | 167 (54%) | 65 (52%) | 102 (56%) | |
| Marital status | | | | 0.014 |
| Never married | 63 (20%) | 16 (13%) | 47 (26%) | |
| Married | 3 (1%) | 2 (2%) | 1 (1%) | |
| Other | 242 (79%) | 108 (86%) | 134 (74%) | |
| Household is indebted | | | | 0.580 |
| Yes | 179 (59%) | 75 (61%) | 104 (58%) | |
| No | 124 (41%) | 48 (39%) | 76 (42%) | |
| **Sex work** | | | | |
| Registered with authorities | | | | 0.569 |
| Yes | 138 (45%) | 54 (43%) | 84 (46%) | |
| No | 170 (55%) | 72 (57%) | 98 (54%) | |
| Clients in typical week (No.) | 6.5 (6.0) | 5.6 (4.2) | 7.1 (6.9) | 0.032 |
| Clients in last 7 days (No.) | 2.6 (4.1) | 1.9 (2.1) | 3.0 (5.0) | 0.014 |
| Sex work income | | | | |
| in last 7 days ('000 CFAF) | 23.1 (52.2) | 19 (24.7) | 25.9 (64.7) | 0.250 |
| average monthly ('000 CFAF) | 124.5 (105.2) | 125.2 (113.1) | 124 (99.8) | 0.928 |
| share of total income (%) | 84 (21) | 85 (20) | 83 (21) | 0.473 |
| Non-sex work income, average monthly ('000 CFAF) | 24.8 (48.8) | 22.2 (36.1) | 26.6 (55.9) | 0.450 |
| Sexual risk-taking (0 = Limits risks, 10 = Likes to take risk) | | | | 0.528 |
| Low (≤1) | 140 (45%) | 53 (42%) | 87 (48%) | |
| High (> 1) | 168 (55%) | 73 (58%) | 95 (52%) | |
| Condom used with last client | | | | |
| Yes | 298 (97%) | 120 (96%) | 178 (98%) | 0.359 |
| No | 9 (3%) | 5 (4%) | 4 (2%) | |
| Share of regular clients in typical week (%) | 72 (30) | 72 (30) | 71 (30) | 0.859 |
| **Randomisation strata** | | | | 0.707 |
| PrEP ever used + High risk-taking | 34 (11%) | 17 (13%) | 17 (9%) | |
| PrEP never used + High risk-taking | 82 (27%) | 32 (25%) | 50 (27%) | |
| PrEP ever used + Low risk-taking | 67 (22%) | 26 (21%) | 41 (23%) | |
| PrEP never used + Low risk-taking | 125 (41%) | 51 (40%) | 74 (41%) | |
| **Overall F-test** | | | | 0.275 |

*Notes:* PrEP, Pre-Exposure Prophylaxis; p, p-value; CFAF, CFAF franc. Table shows mean (SD) for continuous/integer variables and n (%) for binary/categorical variables. p(H0: C = T) means the p-values from two-sided t-tests of equality between control (C) and treatment (T) groups of means and proportions for continuous and binary variables, respectively, and from F-tests for variables with more than 2 categories. See Text A in S1 File for definitions of randomisation strata.

the penultimate client by PrEP referral and PrEP use are not statistically significant (Table I in S1 File). Compared with direct elicitation, control group prevalence of condom use with the last client was estimated to be similar with the color-box method and about 15.5 pp lower with the double list experiment (Table I in S1 File). Using either indirect elicitation method, the estimated RD in condom use by PrEP referral is positive but not significant (Table I in S1 File).

**Table 2. PrEP use prevalence and risk difference by treatment status, at endline.**

| | Control (*n* = 126) | | Treatment (*n* = 182) | | Risk difference (*n* = 308) | | |
|---|---|---|---|---|---|---|---|
| | % | 95% CI | % | 95% CI | pp | 95% CI | *p* |
| PrEP use | 11.1 | [6.2, 17.9] | 45.6 | [38.2, 53.1] | 34.5 | [25.4, 46.3] | < 0.001 |
| Ever used PrEP in past year | 15.9 | [10.0, 23.4] | 54.9 | [47.4, 62.3] | 39.1 | [29.4, 48.7] | < 0.001 |

*Notes:* PrEP, Pre-Exposure Prophylaxis; %, percentage; pp, percentage points, CI, confidence interval; *p*, *p*-value. Percentages of control group (delayed PrEP referral) and treatment group (active PrEP referral) that reported using oral PrEP at the time of the endline interview (PrEP use) and within the 12 months preceding that interview (Ever used PrEP in past year). Risk difference was estimated from logistic regression of PrEP use measure on treatment group indicator. *p*-value is for the test that this difference is zero.

**Table 3. Adjusted risk differences (RDs) in condom use and HIV/STI risk perceptions by PrEP referral and use.**

| | PrEP referral (*n* = 308) | | | PrEP use (*n* = 308) | | | Ever used PrEP in past year (*n* = 308) | | |
|---|---|---|---|---|---|---|---|---|---|
| | RD | 95% CI | *p* | RD | 95% CI | *p* | RD | 95% CI | *p* |
| **Condom used (Yes/ No)** | | | | | | | | | |
| With last client (pp) | 3.3 | [−4.0, 10.6] | 0.376 | 7.9 | [−10.4, 26.3] | 0.397 | 7.4 | [−9.3, 24.0] | 0.386 |
| With all last 3 clients (pp) | 11.0 | [0.8, 21.2] | 0.034 | 25.8 | [5.2, 46.4] | 0.014 | 24.3 | [5.6, 42.9] | 0.011 |
| **HIV risk perceptions (Yes/ No)** | | | | | | | | | |
| High risk without condom (pp) | 4.4 | [−4.4, 13.2] | 0.328 | 10.7 | [−12.4, 33.8] | 0.364 | 10.2 | [−9.1, 29.6] | 0.300 |
| Low risk with condom (pp) | 1.3 | [−1.3, 4.0] | 0.327 | 7.6 | [−27.9, 43.0] | 0.675 | 9.6 | [−19.0, 38.2] | 0.510 |
| High risk without and low risk with condom (pp) | 8.5 | [−2.9, 20.0] | 0.145 | 17.5 | [−13.4, 48.4] | 0.267 | 18.6 | [−8.7, 45.9] | 0.181 |
| **STI risk perceptions (Yes/ No)** | | | | | | | | | |
| High risk without condom (pp) | 0.0 | [−8.9, 9.0] | 0.995 | 1.2 | [−24.8, 27.2] | 0.925 | 1.4 | [−19.9, 22.7] | 0.899 |
| Low risk with condom (pp) | 2.8 | [−7.6, 13.3] | 0.597 | −2.3 | [−38.5, 33.9] | 0.900 | 2.2 | [−27.1, 31.5] | 0.884 |
| High risk without and low risk with condom (pp) | 3.6 | [−7.9, 15.2] | 0.537 | 1.8 | [−32.0, 0.357] | 0.915 | 5.3 | [−24.2, 34.8] | 0.725 |

*Notes:* PrEP, Pre-Exposure Prophylaxis; STI, sexually transmitted infection; RD, risk difference; CI, confidence interval; *p*, *p*-value from test of RD = 0; pp, percentage points. RD by PrEP referral estimated by adjusted logistic regression. RD by PrEP use estimated by recursive bivariate probit with PrEP use instrumented with random assignment to PrEP referral. Adjustment for age, indicator of number of days in last 7 within Ramadan during the 2022 survey interview, marital status in 2020, FSW registration, self-reported sexual risk taking in 2020, prior PrEP experience and the 2020 value of the outcome. Ramadan indicator dropped to achieve convergence of bivariate probit model for Low HIV risk with condom. PrEP use is use at the endline survey. Ever used PrEP in past year is at the endline survey or any time in the preceding 12 months. Risk perceptions were reported on Likert scales and were dichotomised to 1 if above median response and 0 if below. See Table I in S1 File for estimates obtained using alternative linear specifications of these outcomes.

The bottom panel of Table 3 shows estimates of differences by PrEP referral and PrEP use in perceptions of HIV and STI risks when having sex with someone with the respective condition and either using a condom or not. Participants referred for PrEP and those using PrEP were more likely to report that they were (a) very likely to get HIV if they did not use a condom, (b) very unlikely to get HIV if they did use a condom, and (c) both (a) and (b), although none of these estimated RDs are statistically significant. The estimates remain non-significant with a linear (1–5 scale) measure of HIV risk perceptions, although the *p*-values are smaller (Table H in S1 File). There are no significant differences in STI risk perceptions by PrEP referral and PrEP use (Table 3 and Table J in S1 File).

Table 4 shows estimated RDs and MDs by PrEP referral and PrEP use in post-hoc secondary outcomes for risky sexual behaviours. With one exception, the estimates at not statistically significant (at 5%) and they are relatively small in magnitude. The exception is the estimated difference by PrEP use in the probability of perceiving at least one of the last three clients as having a high risk of being HIV–positive. However, this finding is not robust to estimating the respective

**Table 4. Adjusted risk differences (RDs) and mean differences (MDs) in secondary outcomes by PrEP referral and use.**

| | PrEP referral (*n* = 308) | | | PrEP use (*n* = 308) | | | Ever used PrEP in the past year (*n* = 308) | | |
|---|---|---|---|---|---|---|---|---|---|
| | RD/ MD | 95% CI | *p* | RD/ MD | 95% CI | *p* | RD/ MD | 95% CI | *p* |
| **Binary outcomes (Yes/ No)** | | | | | | | | | |
| High sexual risk-taking (pp) | 0.1 | [−11.8, 12.0] | 0.989 | −4.7 | [−37.6, 28.1] | 0.778 | −2.6 | [−30.8, 25.6] | 0.856 |
| High HIV risk of last 3 clients (pp) | 9.6 | [−2.5, 21.8] | 0.120 | 32.0 | [2.2, 61.7] | 0.035 | 21.2 | [−2.3, 44.7] | 0.077 |
| Oral sex with ≥ 1 of last 3 clients (pp) | 0.8 | [−15.4, 17.0] | 0.926 | 1.9 | [−26.3, 30.2] | 0.892 | 0.022 | [−37.5, 41.8] | 0.915 |
| **Non-binary outcomes** | | | | | | | | | |
| Clients in typical week (No.) | −0.004 | [−1.477, 1.469] | 0.996 | −0.012 | [−4.471, 4.447] | 0.996 | −0.011 | [−3.889, 3.868] | 0.996 |
| Average sex acts per client (No.) | −0.029 | [−0.152, 0.094] | 0.645 | −0.101 | [−0.527, 0.324] | 0.641 | −0.076 | [−0.396, 0.244] | 0.642 |
| Share of regular clients in typical week (%) | −2.0 | [−9.1, 5.1] | 0.577 | −5.5 | [−25.2, 14.1] | 0.582 | −5.4 | [−24.5, 13.7] | 0.581 |

*Notes:* PrEP, Pre-Exposure Prophylaxis; RD, risk difference; MD, mean difference; CI, confidence interval; *p*, *p*-value for test of RD/MD = 0; pp, percentage points, No., number; %, percentage. RDs for binary outcomes and MDs for non-binary outcomes. RDs by PrEP referral estimated adjusted logistic regression. RDs by PrEP use estimated by recursive bivariate probit, with random assignment to PrEP referral as instrumental variable for PrEP use. MDs by PrEP referral estimated by adjusted ordinary least squares. MDs by PrEP use estimated by adjusted two-stage least squares, with PrEP referral as instrumental variable for PrEP use. See Table J for estimates with alternative specifications and estimators. Adjustment for age, indicator of number of days in last 7 within Ramadan during the 2022 survey interview, marital status in 2020, FSW registration, self-reported sexual risk taking in 2020, prior PrEP experience and the 2020 value of the outcome. Ramadan indicator dropped in bivariate probit for the oral sex outcome.

MD in a linear measure of this risk perception (Table J in S1 File). Linear measures of the other secondary outcomes are also not associated with PrEP referral and use (Table J in S1 File).

Table 5 shows that only 6% of PrEP users reported believing that they would be *very unlikely* to be infected with HIV if they were to have sex with someone who was HIV–positive and they were neither taking PrEP nor wearing a condom. The percentage describing the risk of HIV infection as *very unlikely* increased to 38% if they were to use PrEP and to 40% if they were to use a condom. And 91% of PrEP users stated that they would be *very unlikely* to become infected if they were to use both PrEP and a condom. Among PrEP users, 20% assessed the risk of HIV infection as *likely* or *very likely* if they were to only use PrEP, compared with only 9% who rated the risk at this level if they were only to use a condom.

## Discussion

This randomised experiment, involving female sex workers (FSWs) in Senegal, contributed to sparse evidence on whether the important health benefits PrEP achieves through reduction in the risk of HIV are partially offset by risk-compensatory behaviour that increases risks of STI that PrEP does not protect against [3]. We did not find evidence that PrEP reduced condom use. Neither referral for PrEP nor its use had a statistically significant effect on the probability of using a condom with the last client. Both PrEP referral and PrEP use were estimated to increase the probability of using a condom with the last three clients. Among the study participants, exposure to PrEP was associated with strengthened perceptions of the effectiveness of condoms in preventing HIV infection, although these estimates are not statistically significant. PrEP exposure was not associated with other measure of risky sex behaviour.

While we focussed on PrEP use, it is important to note that PrEP is not offered in isolation. The way it is implemented may influence outcomes. A PrEP programme provides an additional health system touchpoint to potentially improve sexual health and influence sexual behaviours of a hard-to-reach population, such as sex workers. In Senegal, PrEP users were advised to continue using condoms because PrEP does not protect against other STIs. Free condoms are distributed to registered FSWs. Anecdotally, health ministry officials told the study team that more condoms were dispensed at public health centres after the introduction of PrEP, which is consistent with our finding that PrEP increased condom use. Registration of FSWs in Senegal since 1969 potentially contributed to it. While not implemented yet in Senegal, targeting

**Table 5. Beliefs about likelihood of HIV infection, PrEP users only (*n* = 97).**

| | How likely is it that you would become infected with HIV if you were to have sex with someone who is HIV+ and you used: | | | |
| | Neither PrEP nor condom | Only PrEP | Only condom | Both PrEP and condom |
|---|---|---|---|---|
| Very unlikely | 6 (6%) | 37 (38%) | 39 (40%) | 88 (91%) |
| Unlikely | 0 (0%) | 23 (24%) | 31 (32%) | 5 (5%) |
| Equally likely as unlikely | 0 (0%) | 17 (18%) | 18 (19%) | 1 (1%) |
| Likely | 14 (14%) | 15 (15%) | 7 (7%) | 0 (0%) |
| Very likely | 77 (79%) | 5 (5%) | 2 (2%) | 3 (3%) |

*Notes:* PrEP, Pre-Exposure Prophylaxis. Table shows frequencies (%) of participants who reported currently using PrEP at time of endline survey in both the treatment and control groups.

STI screening on PrEP users may increase the benefits of PrEP. The optimal frequency of this screening would need to be determined [12].

We found no evidence to suggest that PrEP exposure eroded beliefs in the effectiveness of condom use in preventing HIV. While not statistically significant, among study participants, estimates across various specifications consistently suggest that those exposed to PrEP perceived condoms to be more effective. Further, a substantial proportion of PrEP users reported believing that condom use provides substantial additional protection to PrEP, and vice versa. Together with an increase in condom use, these pieces of evidence are consistent with PrEP users currently seeing it as complementary to condom use in HIV prevention, and not a substitute for condoms.

Nonetheless, PrEP is highly effective if taken as prescribed [1–2]. If trust in PrEP were to increase, then there may be a risk of falling condom use in response to adjusted beliefs about the marginal benefit of condoms in preventing HIV. In that scenario, prevalence of STIs would be expected to rise.

Optimism about lower HIV risk due to PrEP rollout may spill over to non-users of PrEP, which would also have consequences for HIV prevalence. Such scenarios suggest that, notwithstanding the findings of this study, vigilance is warranted as beliefs about the effectiveness of PrEP and condoms evolve.

This study had several limitations. First, it was underpowered for precise estimation of effects on the condom use outcomes. A post-hoc power analysis based on the ex-ante MDE revealed that, for PrEP use, there is only 24% power for estimation of the effect on condom use with the last client, but 62% power for the effect on condom use with the last three clients. However, we calculated that there is only a 0.3% chance that the statistically significant estimate of PrEP referral increasing condom use with the last three clients is in the wrong direction (Type S error) [28], which means it is unlikely that the study failed to detect true risk compensation behaviour of reduced condom use in response to PrEP. Nonetheless, the finding that PrEP did not reduce condom use should be confirmed by studies with larger samples or the estimate from this study should be combined with other estimates in a meta-analysis.

Lower-than-expected power was partly due to delayed PrEP rollout that did not start, as scheduled, soon after the 2020 survey. This increased survey and sex work attrition. Two-sided non-compliance was also higher than anticipated. PrEP take-up in the treatment group was much lower than 82% seen in a PrEP demonstration study [20]. Furthermore, while study partners were asked to delay PrEP rollout in the control group until after the endline, this proved hard to implement. Nonetheless, randomised PrEP referral did substantially increase PrEP use, which made it possible to estimate the behavioural response, and we identified it only from the experiment-induced random variation in PrEP use. Null effects on secondary outcomes are consistent with no marked behavioural response to PrEP, which gives greater confidence that the finding of no reduction in condom use was not merely attributable to low power.

Second, the outcomes were primarily self-reported and may differ from actual behaviour due to intentional and unintentional misreporting that could result from social desirability bias and recall bias, respectively. If the degree of misreporting

were to have differed between the treatment and control groups, it would bias estimated effects of PrEP. The bias would be upward if the treatment group were to overreport condom use to a greater extent. To assess this risk, in addition to directly asking participants about their use of condoms, we used two indirect elicitation methods. Although these methods tended to give smaller (and not statistically significant) estimates of increased condom use caused by PrEP referral and use than the estimates obtained via direct elicitation, the main result of no negative effect of PrEP on condom use was robust to the measurement of condom use.

There are two caveats to this robustness. First, the estimate obtained with double list experiment may be unreliable because the method embedded an elicitation experiment within the PrEP experiment. This further reduced statistical power, which is the reason it was not used as the primary outcome measure. Second, the colorbox elicitation method gave estimates of condom use prevalence similar to those obtained with direct elicitation, which suggests that if the latter was prone to misreporting, the former was not effective in reducing the error. Future research should further investigate whether PrEP programmes influence incentives to misreport condom use. In this study, without biomarker outcomes, such as STI diagnoses or the detection of prostate-specific antigen (PSA) and Y-chromosomal DNA, it is not possible to test whether the finding that PrEP did not increase unprotected intercourse is robust to using a more objective measure of that outcome. While PSA and Y-chromosomal DNA have been used to measure unprotected sex, these methods also have limitations, including short detection windows, potential discrepancies between markers [29] and complications, like condom slippage and distinguishing between sex with clients and boyfriends. Logistical, ethical and participant comfort considerations made it impractical to use biomarkers in this study setting.

Third, it was infeasible to estimate effects of PrEP on longer-term outcomes. Even though the overall HIV prevalence in Senegal is low (0.3%), among FSWs it is relatively high (6.6%) [17–18]. Hence, an extended delay of PrEP rollout in a control group would likely be considered unethical. Further, compliance in the treatment arm would likely decrease considerably over time, while contamination in the control arm would increase. Attrition from sex work and the survey would worsen with time. Short-term outcomes could differ from longer-term ones, especially since PrEP users among the study participants seemed to perceive that condom use provides substantial additional HIV protection on top of PrEP, even when PrEP should offer 99% protection against HIV when used consistently. Furthermore, the increasing awareness of clients about the introduction of PrEP in Senegal coupled with the low bargaining power of FSWs may reduce condom use in the longer term [11]. There could also be community-level risk compensation of non-PrEP users when awareness of the prevalence of PrEP usage increases among FSWs [30]. Community-level risk compensation would invalidate RCT estimates unless properly accounted for.

Fourth, participants were not specifically recruited for the RCT and were randomised before attempts were made to contact those assigned to the treatment group for consent, leaving scope for differential study participation between the treatment and control groups, which is a particular concern given a high rate of attrition (49%). However, we showed that treatment assignment had no effect on study participation, we adjusted for covariates and demonstrated robustness of the estimated effects on condom use to this adjustment and to matching the analysis cohort participants to the pre-attrition cohort and the treatment group to the control group in the analysis cohort.

Fifth, the study exerted substantial effort to trace and recruit FSWs in the treatment arm. Some of those recruited might not have been reached, made aware of PrEP and sufficiently convinced of its benefits to agree to take it if there were business-as-usual rollout of the medication [31]. This may have increased the proportion of PrEP users in the treatment arm above what could be expected in normal circumstances, although take-up in the RCT was substantially lower than it was in an earlier demonstration study.

Sixth, sex work is legal in Senegal and the system already has existing public healthcare clinics providing care to registered FSWs. In addition, HIV prevalence in Senegal is relatively low. Therefore, the findings of this study may not necessarily extend to countries with different circumstances.

In summary, this study did not find evidence of risk compensation through reduced condom use in response to PrEP. In fact, the estimated effect on one condom use outcome is positive, suggesting that the PrEP programme may have reached FSWs who otherwise would have had less access to free condoms or health system advice on condom use. Findings suggest that beliefs of condom effectiveness may potentially have increased, and condoms are still viewed as complementary to PrEP usage. Larger studies are required to confirm the findings. Studies on longer term outcomes are also necessary as risk compensation may potentially increase with increased experience of PrEP.

## Supporting information

**S1 Trial Protocol.**
(S1_Trial_Protocol.DOCX)

**S1 Consort Checklist.**
(DOCX)

**S1 SPIRIT Checklist.**
(DOCX)

**S1 File. Supplementary materials. Text A:** Stratified randomisation. **Text B:** List experiment method. **Text C:** Colorbox method. **Table A:** Survey questions used to derive outcomes. **Table B:** Pre-attrition cohort characteristics by treatment status, at baseline (2020). **Table C:** Probability of analysis cohort inclusion by treatment assignment and baseline (2020) characteristics ($n = 500$). **Table D:** Characteristics of analysis cohort in 2020 and at endline (2022) ($n = 308$). **Table E:** Outcome means for control and treatment groups at endline. **Table F:** Mean predicted outcomes under counterfactual of no PrEP use. **Table G:** Unadjusted risk differences (RDs) in condom use by PrEP referral and PrEP use. **Table H:** Adjusted risk differences (RDs) in condom use by PrEP referral and PrEP use—robustness to reweighting analysis cohort and treatment group. **Table I:** Adjusted risk differences (RDs) in condom use by PrEP referral and PrEP use—alternative measurements of condom use. **Table J:** Adjusted mean differences (MDs) in secondary outcomes and HIV/STI risk perceptions by PrEP referral and use—alternative specifications.
(S1_File.DOCX)

**S1 Code. Data and do-file.**
(S1_Code.ZIP)

## Author contributions

**Conceptualisation:** Wally Toh, Aurélia Lépine, Cheikh Tidiane Ndour, Owen O'Donnell.

**Formal analysis:** Wally Toh.

**Funding acquisition:** Aurélia Lépine, Owen O'Donnell.

**Investigation:** Aurélia Lépine, Khady Gueye.

**Project administration:** Aurélia Lépine, Khady Gueye, Mame Mor Fall, Abdou Kouda Diop, El Hadj Alioune Mbaye, Cheikh Tidiane Ndour.

**Supervision:** Aurélia Lépine, Khady Gueye, Abdou Kouda Diop, El Hadj Alioune Mbaye, Cheikh Tidiane Ndour, Owen O'Donnell.

**Writing – original draft:** Wally Toh.

**Writing – review & editing:** Aurélia Lépine, Cheikh Tidiane Ndour, Owen O'Donnell.

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
