## [Editor Report · Decision Letter 0]

24 Jul 2024

Dear Dr Lépine, 

Thank you for submitting your manuscript entitled "Effect of pre-exposure prophylaxis on risky sexual behaviour of female sex workers in Dakar, Senegal: A randomised controlled trial" for consideration by PLOS Medicine.

Your manuscript has now been evaluated by the PLOS Medicine editorial staff and I am writing to let you know that we would like to send your submission out for external peer review.

However, before we can send your manuscript to reviewers, we need you to complete your submission by providing the metadata that is required for full assessment. To this end, please login to Editorial Manager where you will find the paper in the 'Submissions Needing Revisions' folder on your homepage. Please click 'Revise Submission' from the Action Links and complete all additional questions in the submission questionnaire. We also kindly ask that you provide a copy of your trial protocol and a completed CONSORT checklist as supporting information upon resubmission. The documents will be made available to the editors and reviewers.

We have noticed that the pre-experiment power calculation does not currently include the exact numbers as outlined in the protocol. We suggest adding these numbers to the power calculation section so that the calculation is clearer to the reviewers.

Please re-submit your manuscript within two working days, i.e. by Jul 26 2024.

Feel free to email me at atosun@plos.org or us at plosmedicine@plos.org if you have any queries relating to your submission.

Kind regards,

Alexandra Tosun, PhD

Associate Editor

PLOS Medicine

---

## [Decision Letter · Decision Letter 1]

26 Sep 2024

Dear Dr Lépine,

Many thanks for submitting your manuscript "Effect of pre-exposure prophylaxis on risky sexual behaviour of female sex workers in Dakar, Senegal: A randomised controlled trial" (PMEDICINE-D-24-02342R1) to PLOS Medicine. The paper has been reviewed by subject experts and a statistician; their comments are included below and can also be accessed here: [LINK]

As you will see, the reviewers commented on the importance of such a study, but raised serious methodological and reporting concerns. We would like to emphasize that a very careful and thorough revision will be essential to satisfy both the reviewers and the academic editor at re-review.

After discussing the paper with the editorial team and an academic editor with relevant expertise, I'm pleased to invite you to revise the paper in response to the reviewers' comments. We plan to send the revised paper to some or all of the original reviewers, and we cannot provide any guarantees at this stage regarding publication.

We ask that you submit your revision by Oct 17 2024. However, if this deadline is not feasible, please contact me by email, and we can discuss a suitable alternative.

Don't hesitate to contact me directly with any questions (atosun@plos.org). 

Best regards, 

Alexandra 

Alexandra Tosun, PhD 

Associate Editor

PLOS Medicine

atosun@plos.org

Comments from the academic editor:

The topic is both important and timely, also in light of a recent systematic review (Murchu et al., 2022) that highlighted the paucity of data on PreP-associated risk compensation.

The study appears to be at high risk of bias, and the advance over existing knowledge with clear implications for patient care, public policy, or clinical research agendas needs to be made much clearer. 

The very high attrition rate (almost 40%, despite the short follow-up period) needs to be discussed in detail and its implications for the reported results. The self-reported outcome measures on condom use are known to be prone to social desirability bias. There were no biological outcomes related to condom use, such as STIs or vaginal detection of prostate-specific antigen or Y-chromosomal DNA, which have been used in previous studies (see e.g. Giguère et al. 2019). This probably cannot be remedied, but should be discussed in greater detail. 

The apparent change in the outcome measure pointed out by the careful statistical reviewer will be important to address. The results of this study seem to be on the inconclusive side, which is only partially acknowledged by the authors.

Also, the reporting should be substantially improved. For example, it's unclear whether the participants gave informed consent, and the exact timing of the FSW randomization remains unclear (Zelen design?).

Comments from the reviewers: 

Reviewer #1: Thanks for the opportunity to read your manuscript. My role is statistical reviewer, so I have focused on the design, data, and analysis that are presented. I have put general comments first, followed by questions relevant to a specific section of the manuscript (with a page/paragraph reference). 

This RCT examines if referral to PrEP in female sex workers from Senegal leads to changes sexual risk-taking. Participants were recruited from a longitudinal study of FSWs, who were initially recruited from referral from FSWs using public health clinics, or local leaders of FSW groups. Potential participants were those active in sex work and did not have a status of HIV positive recorded. Participants were randomised (3:2 ratio PrEP:control), with active attempts in the PrEP arm to contact participants to see if they were interested in receiving PrEP. Randomisation was done on complete sampling frames, with fixed proportions randomised to either treatment or control. Follow-up data collection was part of the ongoing longitudinal study, with 17% of participants randomised to the PrEP arm not responsive, and 21% not responsive in the control arm. The primary outcomes reported in the manuscript is self-report condom use with clients, the first outcome was condom use with last client, the second is condom use with all of last three clients. The main analyses were intention-to-treat with regards to take-up of PrEP after referral. A range of secondary analyses are considered, including take-up of PrEP at the main follow-up visit. The main analyses used linear models and generalised linear models to estimate treatment effects. A per-protocol estimate is included, using instrumental variable analysis. Participants in the PrEP arm were much more likely to report PrEP use. In the main ITT analysis there was no evidence of treatment effect for condom use with last client, and a modest increase in use of a condom with all of the three last clients. 

Did participants in the ongoing longitudinal survey consent to participating in a randomised trial in the future? If not, was a waiver of consent approval sought for this RCT?

Three primary outcomes are identified in the protocol - condom use in last two clients, condom use in last 5 clients, and subjective perceptions of condom use on a likert scale. In the manuscript, there are two primary outcomes, condom use with last client, and condom use with three last clients, with the Likert outcome moved to a secondary outcome. The online registration also mentions three different modes of capturing condom use with clients. Were the list experiment and newly designed instrument detailed in the online registration used in the end? When was the decision to change the definition of the primary outcomes from the protocol to the version in the manuscript? Was this before or after unblinding? 

Missing data in this study is treated with a complete-case analysis, i.e. assuming that data is 'missing completely at random', and with a relatively large number of participants who missed the follow-up survey. An additional analysis that considers the data to be 'missing at random' (e.g. IPW, MI) should be considered in a revision. 

One of the limitations of the study is that study participants were not specifically recruited for the RCT and are randomised before they are contacted for the study (similar to a Zelen's design or Trial Within Cohort). I would also recommend a sensitivity analysis of a 'missing-not-at-random' scenario for any post-randomisation differential effect of the PrEP contact/visit on follow-up (e.g. controlled imputations with delta adjustment or a tipping point analysis).

Page 7, Paragraph 1. Were the two primary outcomes (last client and last three clients) considered co-primary or multiple outcomes? E.g. was 'success' of the trial considered to be effect in both outcomes, or either outcome? 

Were study staff collecting the follow-up data blinded to the treatment allocation?

Page 8, Paragraph 3. What covariates were used in the adjusted regression models? Are these the ones listed on page 9? Are the same variables used in every analysis reported? What was the rationale for including these covariates?

Could a further explanation be provided about the estimated effect - does the 'outcome probability averaged over the sample' mean that the observed margins were used instead of reference values? 

Page 10, Paragraph 3. I would remove the test of baseline characteristics, as the only way to see a 'significant' difference is by chance (in Table 1 as well). CONSORT makes specific recommendations to not report these. 

Table S1. I would drop the p-values from this table, as the only reason there can be a difference between baseline characteristics is by chance. 

Reviewer #2: The article describes the results of a randomised trial among female sex workers in Senegal testing the effects of (randomised) referral to PrEP services and (resulting) PrEP use on self-reported condom use. The need for this study is clearly justified, and the trial is clearly described and reported. The results clearly show no decline in condom use among FSWs referred to PrEP services or ultimately using PrEP, but also show that this may be linked to misunderstandings about the necessity of concomitant condom use alongside PrEP for maximum PrEP impact on HIV risk. This is an important study and results, adding to our knowledge of real-world PrEP use among FSWs. I have some suggestions for clarifications and edits which would further improve this article, below.

Major comments:

1. There was quite substantial attrition in the trial (both survey and sex work attrition), which I would like to see more clearly highlighted in the article. In the results, where you report the numbers excluded for these reasons, please also present the % excluded for these reasons. In the discussion, instead of mentioning the high attrition rate and consequent reduced power at the end of a point about future studies, I would like to see the high attrition listed explicitly as a limitation at the start of one of these points.

2. P20, paragraph 4: 'Table 4 shows estimates of effects of PrEP referral and PrEP use on beliefs about the effectiveness of condoms in preventing HIV and STIs.' Should this be Table 3? Please check. Similarly, later in that paragraph where you refer to SM Table S5, I think this should be SM Table S4. At the end of the paragraph, where you mention 'Estimates of effects of PrEP referral on beliefs about the effectiveness of condoms in preventing other STIs do not show the same consistency, have much larger p-values and much smaller effects', please add references to the relevant tables showing this information (Table 3 and Table S4)

3. P21, paragraph 2: '9% of PrEP users answered that they are likely or very likely to being infected by HIV when using PrEP only versus 20% for condom only' I think these are labelled the wrong way round, so it should be '9%... when using condoms only versus 20% for PrEP only'. Please check and correct.

Minor comments:

1. There are minor spelling and grammatical errors throughout. A few are listed below, but I suggest a thorough proof-reading of the article before resubmission

a. Abstract: Background: generates negative -> generate negative

b. P14, Methods: 'and full protocol is available on the UCL repository' -> and the full protocol is available in the UCL repository

c. P14, Study design and participants: 'excluding those who (a)…' there is no corresponding '(b)'

d. P15, Intervention: 'Those who eligible' -> 'Those who were eligible'

2. Abstract, Findings: for consistency, please include the p-value for final set of results presented (effect of PrEP use on condom use with all 3 last clients), and please include 'pp' as units for this

3. You mention several times that midwives recruited participants - was it really midwives rather than nurses? Midwives assist women in childbirth, so it seems unlikely that they would have a recruitment role in a PrEP trial

4. P16, Outcomes: for the scale eliciting beliefs about the likelihood of contracting HIV/STI, it is unclear how the information in a) (with and without use of a condom) differs from the information obtained from b) and c) together. It may be helpful to reproduce the questions in the appendix?

5. P16, Pre-experiment power calculation: the reader may not know what list experiments are. These are explained further in the discussion, but it may be helpful to clarify here, or add some information in an appendix which can be referred to here

6. P19/p20: is 'recently used PrEP' the same as 'having used PrEP in 2021/22'? If so it would help to define recently used PrEP as this, or even to use the same phrase throughout

7. P20, paragraph 2: 'Table 2 shows estimates of ITT effects of PrEP referral, effects of PrEP use on condom use and effects of having used PrEP in 2021/2022.' Are the effects all on condom use? It's a bit unclear from this sentence.

8. P20, paragraph 3: 'alternative outcome specification' - it would helpful to briefly summarise what this was

9. P21, Discussion, paragraph 1: '…among PrEP users and FSWs recently exposed…' for clarity, I suggest changing this to '…among PrEP users or among FSWs recently exposed'

10. P21, Discussion, paragraph 1: 'When taking all three clients together' I suggest clarifying to say '…all three most recent clients together'

11. P23, last paragraph: 'The latter method…' do you mean the colorbox method? Please make clear.

12. References: reference 1 gives an out-of-date link. For preference, use a peer-reviewed article as a source for this information.

Reviewer #3: The study examines whether oral PrEP use leads to risk compensation among female sex workers (FSWs) in Dakar, Senegal, through a randomized controlled trial. The intervention arm received a 4-month PrEP referral program, while the control arm did not receive PrEP referral until the end of the study. The authors found that PrEP use did not significantly decrease condom use or increase other sexual risk behaviors, suggesting no immediate risk compensation effects. While the results are reassuring, there are several methodological areas that could be improved, particularly how the planned intention-to treat analysis was handled. Major critiques are highlighted below:

Introduction

1. The PrEP landscape is rapidly evolving with multiple PrEP options. Please consistently specify the modality (oral) of the PrEP referenced. 

Methods

1 The authors mention that participants gave consent to each survey, but it is not clear if the participants also provided consent to the RCT?

2 Randomization was stratified by PrEP experience and sexual risk-taking behavior, which makes sense. Can the author briefly state their rationale in the text?

3 It is not clear when the randomization occurred? Please clarify. 

4 Additional details of randomization process are needed. Who was blinded to groups assignment? Were the staff who administered the endline survey blinded? Where the staff who contacted the participant in the intervention arm blinded? Or what measures were put in place to minimize bias in assessment?

5 The intervention only lasted for 4 months, from October 2021 to January 2022, while there are two years between the two waves of the survey. HIV risk and exposure is a dynamic and not static process that there can be a lot of changes even in a space of 3 months. Can the author clarify their rationale for conducting only a 4-month PrEP referral program during the two-year follow up period?

6 Additional information about the intervention is needed? How were the midwives and FSW facilitators trained in referring FSWs to PrEP? How often did they reach out, and in what way? If a participant showed no interest in PrEP, would they receive another contact? How long did it typically take from referral to PrEP screening?

7 For data collection, can the author add more details about the variables collected? For example, how was PrEP utilization assessed in the survey?

8 The authors examined PrEP use in two ways: current use at the time of the endline survey and recent PrEP use but there is no clear description of this classification is provided. Can the authors define recent PrEP use? Based on the results section, recent PrEP use refers to PrEP use in 2021 and 2022. Can the author justify why this period was used? Why was PrEP use during the entire follow-up period or after the start of the PrEP referral program not measured?

9 Please clarify what "exposure to Ramadan within 7 days" refers to?

10 Can the authors specify more details on the inclusion and exclusion criteria for the actual analysis?

11 The intervention was referral for PrEP. So, there was no attempt for any form baseline assessment for the control group. A design in which the evaluation between the two groups is not similar always bound to introduce bias. Please justify. 

12 The RCT was designed to evaluate the effect of the 4-month PrEP referral program on changes in sexual behaviors among FSWs. However, the question the presented in analysis presented by authors aimed to answer focuses on PrEP use and sexual behaviors, which is not the primary question of the study. 

13 In an intent-to-treat analysis, participants are analyzed based on the group they were randomized to regardless of whether they actually participate or not in the assigned group or not. So, the primary which focus on 182 vs 126, is modified ITT

14 For the FSW population, as the probability of condomless sex is already high. Did authors consider focus on analysis focused change in trends or change from baseline analysis? 

15 Similar to my comment above, this is a longitudinal study but there was no attempt assess the change in risk behaviors, given that the main question is about risk compensation (i.e., change in risk)?

16 It seems the analytic implicitly assumes that the covariance of the measured matrices to be homogeneous in the two groups NOT heterogeneous which seems be an oversimplification given the differences in outcome assessment for treatment vs control. Indeed, if there is heterogeneity in variance across groups, repeated measures analysis utilizing Kenward and Roger's adjustment would be preferred. Please provide justification for your approach.

17 In the power estimation, an attention rate of 10% was assumed, However, from the n=500 randomized individual to final analytic sample n=308, it represents nearly 40% attrition and for the true ITT sample n=415, it's ~17% attrition. It possible that the observed null effect could be due to type II error? Can the authors do power simulation based in the final sample size to estimate the available power? Also, what were the characteristics of those no included in those two samples compared to those not included in the analysis?

Results

1. Table 1 should present the characteristics of participants who were randomized n=500.

2. It is not appropriate to call an analysis which excludes participants who indicated that they no longer engaged in sex work in the 2022 survey post-randomization an intent-to-treat analysis. Importantly, the action of no longer being engaged in sex work, is an outcome of the study—as it may indicate is the change HIV and thus no need for PrEP. Thus, an ITT analysis should focus on n=247 in the intervention arm vs n=158 in control arm. The arbitrary division of analysis into no longer in sex work vs still sex work non-responders leads to destruction of information and unjustified causal association interpretation. Can the author conduct an analysis by including these participants?

3. In the second paragraph of the results section, the first sentence states that randomization occurred two years after the 2020 survey. Clarify when the randomization actually took place?

4. HIV risk as well as perceived risk is a dynamic process with change from time to time. Thus, to use data collected 2 years prior as the baseline may be an oversimplification of the constancy assumption. Although on average randomization would be expected to balance the group on measured/unmeasured factors, the sample size of 500 with 3:2 ratio is quite small to guarantee balance between groups especially within the key stratification categories. 

5. What was the median time from randomization to the administration of the endline survey? How did that differ by arm. In the intervention arm, what was the median time large from referral to picking up PrEP?

6. What proportion of participants in the control arm had PrEP referral and/or PrEP exposure during the intervention period? And how did that affect estimates

7. The paragraph starting with "Table 4 shows estimates of effects of PrEP referral" should refer to Table 3 instead.

8. In Tables 2 and 3, the reference group means for the regressions of PrEP referral, PrEP use, and ever PrEP use are different. Please specify the reference group means separately.

Discussion

1. Can the author discuss potential contamination of the control group and its implications. 

Minor 

1. Third paragraph in the section of study design and participants, methods, delete "a)" 

2. 2nd paragraph in the section of intervention, methods, it should be "experimental and control arms"

Reviewer #4: Important and interesting RCT looking at whether prep use causes risk compensation. This is important for understanding the full benefits of prep. The results are interesting although the high LTFU rate raised concerns, and it is a pity that the study relies on self-reported condom use which is a weak measure. My comments are as follows:

1. Great introduction to the paper laying out the existing knowledge.

2. It was worrying to see the authors had limited data on aspects of the prep intervention, although it is unclear. Can you make it clearer what data was available for each arm and what was not. 

3. They say they checked self-reported prep use data - how well could that be done?

4. It is a pity the authors could not use a biological test or an anonymous method for condom use on the FSW because self-reported condom use is biased. This is something the study authors recognise and may have biased the prep users more if they were attending services more than the control arm. They note lower condom use in one of the anonymous methods - I assume they could not compare between prep and non prep users. 

5. LTFU rates were high (only about 60% were surveyed at follow up_- what could the effect of that be? Did it occur more in the unregistered FSWs. Is it possible that those with higher LTFU had more indebtedness and so may have been more willing to use condom less while on PrEP. 

6. I notice condom use decreased over the prep period by 10pp potentially due to decreases in condom use. Could there also be population optimism about HIV that results in decreases in prep users and non-users. 

7. Prep use can be very transient in FSWs - what did the authors find in terms of retention on PrEP. Is it possible to get this data from their survey or the providers? I assume only long term users would think about reducing condom use, so maybe the follow up here is too short?

8. Did an individuals perceived unlikelihood of getting HIV with Prep predict lower condom use in FSWs on PREP? 

9. The authors state 'Anecdotally, a notable reduction in condom supplies of the MoH was observed after the introduction of PrEP, consistent with the increase in condom use found in the study.' Should it say decrease in condom use found in the study.

---

* Please upload any figures associated with your paper as individual TIF or EPS files with 300dpi resolution at resubmission; please read our figure guidelines for more information on our requirements: http://journals.plos.org/plosmedicine/s/figures. While revising your submission, please upload your figure files to the PACE digital diagnostic tool, https://pacev2.apexcovantage.com/. PACE helps ensure that figures meet PLOS requirements. To use PACE, you must first register as a user. Then, login and navigate to the UPLOAD tab, where you will find detailed instructions on how to use the tool. If you encounter any issues or have any questions when using PACE, please email us at PLOSMedicine@plos.org.

* FINANCIAL DISCLOSURES: The funding statement should include: specific grant numbers, initials of authors who received each award, URLs to sponsors’ websites. Also, please state whether any sponsors or funders (other than the named authors) played any role in study design, data collection and analysis, the decision to publish, or preparation of the manuscript. If they had no role in the research, include this sentence: “The funders had no role in study design, data collection and analysis, decision to publish, or preparation of the manuscript.”

* COMPETING INTEREST: All authors must declare their relevant competing interests per the PLOS policy, which can be seen here: https://journals.plos.org/plosmedicine/s/competing-interests

For authors with ties to industry, please indicate whether any of the interests has a financial stake in the results of the current study.

* DATA AVAILABILITY: The Data Availability Statement (DAS) requires revision. Since you have indicated that the data are accessible in the UCL repository database, please include the DOI or accession number.

* ETHICS: Please see the reviewer's comments regarding consent and clarify. Also, please indicate whether informed consent was written or oral.

FIGURES AND TABLES

SUPPLEMENTARY MATERIAL

REFERENCES

* Where website addresses are cited, please include the complete URL and specify the date of access (e.g. [accessed: 12/06/2024]).

STUDY TYPE-SPECIFIC REQUESTS

* PLOS Medicine requires that all trials be prospectively registered in one of registries recognized by WHO. Please ensure that study registration details are included in the Methods section.

* Please structure the Methods section using the following sub-headings: Study design and participants, Randomization and masking, Procedures, Outcomes, Statistical analysis.

* The following outcomes measures appear to differ between the submitted manuscript and the protocol [and/or trial registry]. Please clarify and explain all discrepancies between the paper and protocol.

1) Primary outcome: Condom use with each of the last two clients and the last five clients measured via a combination of these methods: Via direct questioning, Via list experiment, Via a newly-designed instrument aimed at eliciting individual condom use without 

2) Primary outcome: Subjective perceptions of the necessity of condom use measured using a Likert-like scale

3) Secondary outcome: Price of each of the last two clients

4) Secondary outcome: Earnings from sex work in a fixed time frame, e.g. 30 days

5) Secondary outcome: Household expenditure in a fixed time frame, e.g. 30 days

6) Secondary outcome: Food insecurity

7) Secondary outcome: Self-reported STI symptoms with last two clients

8) Secondary outcome: Mental health measured using the PHQ-9 questionnaire

* If the outcomes were not prespecified in the protocol, please define them in the Methods (Outcomes section) as post hoc and explain why they were added. Post-hoc comparisons should be presented as hypothesis generating rather than conclusive.

* Please ensure that all prespecified outcomes (primary, secondary, and exploratory) are listed in the Methods/Outcomes section and indicate whether there are outcomes that are not presented in the current report.

* Please specify the dates (Month Day, Year) during which study enrollment and follow up occurred.

* Please include absolute numbers wherever you report percentages; eg, n/N (%)

* Please present the safety data for the study including numbers of specific events and whether or not adverse events are thought to be related to treatment. AEs should be reported in the abstract, per CONSORT and CONSORT-Harms.

* Please complete the CONSORT checklist (https://www.equator-network.org/reporting-guidelines/consort/) and ensure that all components of CONSORT are present in the manuscript, including how randomization was performed, allocation concealment, blinding of intervention, definition of lost to follow-up, power statement. When completing the checklist, please use section and paragraph numbers, rather than page numbers.

* Please report your abstract according to CONSORT for abstracts, following the PLOS Medicine abstract structure (Background, Methods and Findings, Conclusions) https://www.equator-network.org/reporting-guidelines/consort-abstracts/

* If your trial had to undergo important modifications in response to extenuating circumstances, please complete the CONSERVE-CONSORT checklist and provide in your Supporting Information; (https://www.equator-network.org/reporting-guidelines/guidelines-for-reporting-trial-protocols-and-completed-trials-modified-due-to-the-covid-19-pandemic-and-other-extenuating-circumstances-the-conserve-2021-statement/). When completing the checklist, please use section and paragraph numbers, rather than page numbers.

* In keeping with our commitment to Open Science, please include the study protocol document and analysis plan (including any amendments) as Supporting Information to be published with the manuscript if accepted.

* Please note that PLOS Medicine requires prospective, public registration of a data sharing plan (as part of mandatory clinical trials registration) for all clinical trials that began enrollment on or after January 1, 2019, in accordance with ICMJE requirements.

---

## [Decision Letter · Decision Letter 2]

2 Apr 2025

Dear Dr. Lépine,

Thank you very much for re-submitting your manuscript "Effect of pre-exposure prophylaxis on risky sexual behaviour of female sex workers in Dakar, Senegal: A randomised controlled trial" (PMEDICINE-D-24-02342R2) for review by PLOS Medicine.

Thank you for your detailed response to the reviewers' and editors' comments. I have discussed the paper with my colleagues, and it has also been seen again by all of the original reviewers. The changes made to the paper were mostly satisfactory to the reviewers. As such, we intend to accept the paper for publication, pending your attention to the reviewers' and editors' comments below in a further revision. When submitting your revised paper, please once again include a detailed point-by-point response to the reviewers' and editorial comments.

[LINK]

In revising the manuscript for further consideration here, please ensure you address the specific points made by each reviewer and the editors. In your rebuttal letter you should indicate your response to the reviewers' and editors' comments and the changes you have made in the manuscript. Please submit a clean version of the paper as the main article file. A version with changes marked must also be uploaded as a marked up manuscript file. Please also check the guidelines for revised papers at http://journals.plos.org/plosmedicine/s/revising-your-manuscript for any that apply to your paper.

We ask that you submit your revision within 1 week (Apr 09 2025). However, if this deadline is not feasible, please contact me by email, and we can discuss a suitable alternative.

Please do not hesitate to contact me directly with any questions (atosun@plos.org). If you reply directly to this message, please be sure to 'Reply All' so your message comes directly to my inbox.

We look forward to receiving the revised manuscript.

Sincerely,

Alexandra Tosun, PhD

Associate Editor

PLOS Medicine

plosmedicine.org

Comments from Reviewers:

Reviewer #1: Thanks for the revised manuscript and responses to my original review. The responses and updates to the manuscript have resolved queries from my original review.

Changes to the primary outcome were made during the trial but before unblinding. A reasonable justification is given for this (from my perspective as a non-content expert), and a set of sensitivity analyses are completed that are consistent with the main results. 

A sensitivity analyses weighting data with entropy weights is provided, weighting follow-up to baseline data, the results are consistent with the main results. 

Reviewer #2: The authors have addressed all of my comments satisfactorily.

Reviewer #3: I appreciate the efforts that the authors took to address my comments. I do not have any follow up questions. Great job! 

Reviewer #4: The author has thoughtfully responded to my comments. My only remaining comments are that it would be good to say how they added text in response to my comment 3 (about checking self reported data), and i still think the following sentence is not clear '"Anecdotally, a notable reduction in condom supplies of the MoH was observed after the introduction of PrEP, consistent with the increase in condom use among PrEP users found in the study'. Do they mean the MOH stocks of condoms reduced because of increased use of condom, or did they supply less each month but condom use still went up?

[LINK]

Requests from Editors:

GENERAL

* In trials, there is usually a distinction in the language in terms of causal vs associational for primary and secondary trial outcomes. It would be beneficial to use associational language in the discussion and other sections for secondary outcomes.

* Please ensure that all abbreviations are defined at first use throughout the text (including statistical abbreviations). Please also check figures and tables.

* Please review your text for claims of novelty or primacy (e.g. 'for the first time', ‘novel’) and remove this language.

* Please check that any use of statistical terms (such as trend or significant) are supported by the data, and if not please remove them.

* Please ensure that tables and figures, including those in supplementary files, are appropriately referenced in the main text.

* Statistical reporting: Please revise throughout the manuscript, including tables and figures.

a) Please report statistical information as follows to improve clarity for the reader "22% (95% CI [13%,28%]; p</=)". 

b) Please separate upper and lower bounds with commas instead of hyphens as the latter can be confused with reporting of negative values. 

c) Please define statistical definitions at first use and repeat the abbreviated definitions (HR, CI etc.) for each set of parentheses.

* Please avoid referring to the study participants as sample. Please re-place with cohort, participants or similar.

* We have noticed that in some tables the risk difference/effect estimate column is labeled as one or the other. Please review and use a consistent format.

* When presenting results, we suggest not repeating numerical results with range and/or intervals in the main text if these results can also be found in the corresponding tables and figures. We think that this would streamline the text.

* In general, we feel that the main text, especially the results section, is not always easy to follow. Please revise carefully for clarity and conciseness, and be sure to guide the reader through your results.

DATA AVAILABILITY

* Please include the DOI or accession number.

ABSTRACT

* Please confirm that your abstract complies with our requirements, including providing all the information relevant to this study type https://journals.plos.org/plosmedicine/s/submission-guidelines#loc-abstract

* Per CONSORT, please note that only the primary outcome of the trial should be reported in your Abstract. Secondary outcomes should only be included in the Abstract if all secondary outcomes are fully reported. For trials that have many secondary outcomes, the Abstract should be limited to reporting the primary outcome.

* Please ensure that all numbers presented in the abstract are present and identical to numbers presented in the main manuscript text.

* On line 44, we suggest writing ‘T’ and ‘C’ in full. 

* ll.46ff: Because you are describing percentage points, which describe a difference between two percentages, the estimated effect should be either increasing or decreasing. Please rephrase and be sure to indicate if the results are not significant.

* Please note that you switch between "control group mean" and "reference mean". Do you always refer to the control group? If so, we recommend using the same description throughout.

* We have noted that you discuss short-term effects, but have not provided details on time points or time lines between referral and self-reported outcomes. What was the average time between referral and endline survey?

* l.50: "There were no notable effects on other risky behaviours." - since this statement refers to the secondary outcomes, we suggest using associational language (if not removed, see second comment under Abstract).

AUTHOR SUMMARY

* We suggest changing the second bullet point to (split into two):

1) Risk compensation may be more pronounced among female sex workers (FSWs), as condomless sex is better remunerated. 

2) According to a literature search in January 2024, there is a lack of randomised controlled trials testing whether PrEP uptake impacts the prevalence of unprotected sex among FSWs. For men who have sex with men, the evidence was mixed, with more recent studies showing risk compensation.

* We feel that the statement, “To our knowledge, this study provides the first evidence from a randomised experiment on whether PrEP causes compensatory risky sexual behaviour by FSWs.”, is somewhat repetitive of the statement on the literature search. We suggest removing it.

* l.92: We suggest removing the word ‘strong’.

* l.93: Again, we note that you discuss short-term effects, but have not provided details on time points or time lines between referral and self-reported outcomes.

* ll.93-95: “The data are consistent with users of oral PrEP believing that combined protection with condoms is more effective than using PrEP alone.” – as written, it is not clear whether this statement reflects another finding of your study or whether it is based on recent literature.

* l.98: “It is generally believed…” – is this based on scientific literature or scientific discourse among the research community?

* ll.100-103: We suggest creating a new bullet point for the statements starting “The evidence...” and ending “…in other setting”.

* ll.104-107: Are there any indicators that the belief on combined PrEP and condom use is changing? We are not sure whether this statement is really needed here.

* ll.107-109, we suggest changing to: Studies with longer follow-up are needed to evaluate behavioral responses. The study also demonstrates the difficulties of conducting a randomized controlled trial in this setting.

* Please note that the last bullet point should clearly state the main limitation(s).

INTRODUCTION

* Please change the heading to ‘Introduction’.

METHODS AND RESULTS 

* Please provide the approval numbers from both ethics committees. 

* l.284: “sample of 500” – please change to: “We assumed a drop-out rate of 10% from the original study size of 500 individuals due to survey and sex work attrition,…”. 

* ll.351-353: Please remove the ‘Role of the funding source’ statement from the main text. These details should only be included in the metadata in the online submission form.

* ll.336-337: “number of days in the seven preceding the endline interview that were within Ramadan” – please briefly explain the reason for adjusting for days within Ramadan.

* Figure 1: Please define ‘PrEP’.

* l.367: We suggest including the total n-number here for clarity (n=308).

* l.382ff: Please clarify that these results refer to the total cohort.

* ll.387-392: In what table/figure can readers find these values? Please provide a reference.

Throughout, we feel it needs to be more clear when results are not significant (cross the null).

* l.394ff: We feel that it may be confusing to readers that you present risk differences (RD) in Table 2 while you continue to refer to "effect estimates" or "estimated effect" in the text. Also, in the table you present the risk difference as a decimal number, whereas in the text you present it as a percentage point. We feel that a consistent format (between tables/figures and text) would be preferable for the reader.

* l.397ff: “The point estimate corresponds to a 3.8% increase over the reference (control) group mean (84.9%)” – We are not sure whether we missed it, but have you provided the outcome mean values for the intervention group anywhere (equal to S5 Table for the reference group)? 

* l.424ff: “The signs of the point estimates are consistent with PrEP referral and use both strengthening beliefs that use of a condom is effective in preventing HIV, although the 95% confidence intervals all include zero.” – We think that this statement is not 100% clear and that the interpretation of the direction of risk differences should not be part of the results section.

* l.431: “using the full range of perceived HIV risks” – please re-phrase and be more specific.

* l.431: "The direction of these differences is robust..." - although the linear interpretation yielded similar directions, the results were not significant, i.e. we would suggest toning down these findings.

* Tables: Please ensure to define all abbreviations used in tables and their descriptions, such as ‘PrEP’, ‘STI’, ‘CI’.

* Table 1: Is there a reason why you decided to report Share of XYZ as decimal numbers instead of percentages?

* Table 2: In the figure description, you write “See S4 Table for estimates obtained using all values of these outcomes”. We are not 100% sure what the sentence means and also noted that S4 Table shows the characteristics of analysis cohort in 2020 and at endline (2022). Please clarify.

* Table 3: “RDs were estimated as explained in notes to Table 2.” – please note that tables (and figures) should be self-explanatory on a standalone basis. Please repeat the explanation.

* Table 3: “See S5 Table for estimates with alternative specifications and estimators” – please check whether this is a correct reference. 

DISCUSSION

* Please change the ‘Conclusions’ heading to ‘Discussion’.

* Please remove any subheadings in the Discussion, including the Conclusion subheading.

* l.491: Please note that in the discussion new results should not be presented. Please present the results on associations of risky sexual behaviours with PrEP use to the Results section (including the reference to S10 Table).

General Editorial Requests

---

## [Editor Report · Decision Letter 3]

30 Apr 2025

Dear Dr. Lépine,

Thank you very much for re-submitting your manuscript "Effect of pre-exposure prophylaxis on risky sexual behaviour of female sex workers in Dakar, Senegal: A randomised controlled trial" (PMEDICINE-D-24-02342R3) for review by PLOS Medicine.

Thank you for your detailed response to the reviewers' and editor’s comments. We think the Results section has been greatly improved. I have discussed the paper with my colleagues, and there are a few minor editorial issues that need to be addressed before we can accept the manuscript for publication. You will see that there are several editorial points pertaining to the changes in the trial protocol that require your attention. When submitting your revised paper, please once again include a detailed point-by-point response to the editorial comments. Please revise the paper accordingly, and submit the final revision within 1 week (May 07).

In revising the manuscript for further consideration here, please ensure you address the specific points made by the editors. In your rebuttal letter you should indicate your response to the editors' comments and the changes you have made in the manuscript. Please submit a clean version of the paper as the main article file. A version with changes marked must also be uploaded as a marked up manuscript file. Please also check the guidelines for revised papers at http://journals.plos.org/plosmedicine/s/revising-your-manuscript for any that apply to your paper.

A reminder that when your manuscript is accepted, an uncorrected proof of your manuscript will be published online ahead of the final version, unless you've already opted out via the online submission form. If, for any reason, you do not want an earlier version of your manuscript published online or are unsure if you have already indicated as such, please let the journal staff know immediately at plosmedicine@plos.org.

Please do not hesitate to contact us directly with any questions (atosun@plos.org). If you reply directly to this message, please be sure to 'Reply All' so your message comes directly to our inbox.

We look forward to receiving the revised manuscript.

Sincerely,

Alexandra Tosun, PhD

Associate Editor 

PLOS Medicine

plosmedicine.org

Requests from Editors:

TRIAL PROTOCOL

*Please upload the most recent version of the protocol as a Supporting Information file.

* Please note that you will need to provide a list of amendments as a Supporting Information file. Please confirm that these changes have been approved by an IRB and confirm that the approval was given prior to the data lock.

* We also need you to transparently explain in the Methods that the original protocol said X for primary outcomes, that the decision was made to change to Y, and that the changes were made prior to data lock. Please clarify if the changes were made before any survey questions were asked. As you have already done, please remember that it is important to direct readers to where you conducted additional analyses to show that the changes you made to your original plan did not substantially change the conclusions.

* For reporting transparency, we encourage you to provide the 2025 SPIRIT checklist and ask you to update the CONSORT checklist to 2025 CONSORT. You can find the checklists here: https://www.consort-spirit.org/

Other editorial comments:

*Abstract, line 45, “308 (61.6%) were used for analysis”: We suggest changing to, “308 (61.6%) were included in the analysis”.

*Abstract: Thank you for your clarification regarding the statement, " Estimated effects of PrEP referral and PrEP use on condom use with the last client were positive but not statistically significantly different from zero." We agree with the presentation of the results. If you wish, you may add the numerical results.

*Abstract: Please indicate that the analysis is a modified intention to treat (ITT) analysis.

*Please ensure to add a reference to the CONSORT checklist, the trial protocols and the list of amendments in the main text.

*Data Availability: Please note that we can only publish the manuscript with the final DOI or accession number.

---

## [Editor Report · Decision Letter 4]

20 Jun 2025

Dear Dr Lépine, 

On behalf of my colleagues and the Academic Editor, Matthias Egger, I am pleased to inform you that we have agreed to publish your manuscript "Effect of pre-exposure prophylaxis on risky sexual behaviour of female sex workers in Dakar, Senegal: A randomised controlled trial" (PMEDICINE-D-24-02342R4) in PLOS Medicine.

I appreciate your thorough responses to the comments from the reviewers and editors throughout the editorial process. Thank you for addressing our final concerns via email. We look forward to publishing your manuscript. As you know, editorially, there are a few points that still need to be addressed before publication. Please implement the final points in your manuscript as you addressed them in your email. We have outlined the requests once more below. We will carefully check whether the changes have been made. If you have any questions or concerns regarding these final requests, please feel free to contact me at atosun@plos.org.

Please see below the minor points that we request you respond to:

1) According to the trial protocol you have provided, some of the post-hoc secondary outcomes seem to overlap with the pre-specified secondary outcomes, including 'Number of occasional and regular clients seen in a fixed time frame, e.g. 7 days', 'Perceived HIV risk of clients', 'Type of sex acts with last two clients'. Are the post-hoc secondary outcomes defined as such because they focus on the last three clients, which deviates from the original protocol?

2) Please clarify whether the Zelen design was approved by the IRB as its design usually requires a waiver of the requirement for informed consent prior to random assignment of treatment.

3) Please provide the exact dates of randomization (beginning and end) and the exact dates of consenting (as the end date of consenting is ~equivalent to the last date of enrolment) in the Abstract and the Methods section.

4) Please note that although the protocol states "references," there are none.

5) In the Methods section of the manuscript, please clarify that you had permission to randomize the participants of the 2020 survey and who gave you that permission. Also, please clarify whether the participants were informed in 2020 that they might be contacted again in the future.

6) Data Availability: Thank you for providing the link. We understand that the link is still inactive at the moment. Please note that we can only publish the manuscript if it does not have a final, active DOI or accession number.

Before your manuscript can be formally accepted you will need to complete some formatting changes, which you will receive in a follow up email (including the editorial requests above). Please be aware that it may take several days for you to receive this email; during this time no action is required by you. Once you have received these formatting requests, please note that your manuscript will not be scheduled for publication until you have made the required changes.

PRESS

Sincerely, 

Alexandra Tosun, PhD 

Senior Editor 

PLOS Medicine